# Concordance among Swedish, German, Danish, and UK EQ-5D-3L Value Sets: Analyses of Patient-Reported Outcomes in the Swedish Hip Arthroplasty Register

**DOI:** 10.3390/jcm10184205

**Published:** 2021-09-17

**Authors:** Fitsum Sebsibe Teni, Ola Rolfson, Jenny Berg, Reiner Leidl, Kristina Burström

**Affiliations:** 1Health Outcomes and Economic Evaluation Research Group, Department of Learning Informatics, Management and Ethics (LIME), Stockholm Centre for Healthcare Ethics, Karolinska Institutet, 177 71 Stockholm, Sweden; ola.rolfson@vgregion.se (O.R.); jenny.berg@ki.se (J.B.); kristina.burstrom@ki.se (K.B.); 2Department of Orthopaedics, Institute of Clinical Sciences, Sahlgrenska Academy, University of Gothenburg, 413 45 Gothenburg, Sweden; 3Swedish Hip Arthroplasty Register, 413 45 Gothenburg, Sweden; 4German Research Center for Environmental Health, Institute for Health Economics and Health Care Management, Helmholtz Zentrum München, 857 64 Neuherberg, Germany; leidl@helmholtz-muenchen.de; 5Munich Center of Health Sciences, Ludwig-Maximilians University, 805 39 Munich, Germany; 6Equity and Health Policy Research Group, Department of Public Health Sciences, Karolinska Institutet, 177 71 Stockholm, Sweden

**Keywords:** EQ-5D-3L, experience-based perspective, hypothetical perspective, total hip replacement (THR), time trade-off (TTO), valuation, value sets, visual analogue scale (VAS)

## Abstract

**Background**: Application of different value sets to health-related quality of life (HRQoL) measured with the EQ-5D-3L may lead to different results due to differences in methods, perspectives, and countries used. Focusing on concordance, this study aimed at understanding the implications of applying EQ-5D-3L value sets from Sweden, Germany, Denmark, and the UK to evaluate HRQoL of patients undergoing total hip replacement (THR) in Sweden before and after surgery. **Methods**: We performed a longitudinal study of patients in the Swedish Hip Arthroplasty Register from preoperative stage to 1-year follow-up (*n* = 73,523) using data collected from 2008 to 2016. Eight EQ-5D-3L value sets from the four countries were compared based on a valuation method (visual analogue scale (VAS) or time trade-off (TTO)), perspective (experience-based or hypothetical), and country. Concordance among the value sets with patient-reported EQ VAS score was also assessed. Longitudinal changes in EQ-5D-3L index over the 1-year follow-up were compared across value sets by method, perspective, and country. **Results**: Value sets based on the same method and perspective showed higher concordance in EQ-5D-3L index at both measurement time points than other comparisons. In the comparisons by perspective, VAS value sets showed higher concordance than TTO value sets. The Swedish VAS and the Danish TTO value sets showed the highest levels of concordance with patient-reported EQ VAS scores. Generally, value sets based on the same method and perspective had the smallest mean differences between changes in EQ-5D-3L indices from preoperative to 1-year postoperative follow-up. **Conclusion**: Among THR patients value sets based on the same method and perspective, a direct transfer of results across countries could be meaningful. In cases of differences in methods and perspectives among value sets, transfer of value sets across settings would have to consider conversion through crosswalk.

## 1. Introduction

EQ-5D is among the most commonly employed generic health-related quality of life (HRQoL) instruments validated across diverse settings [1]. It contains a five dimensional descriptive system comprising mobility, self-care, usual activities, pain/discomfort, and anxiety/depression, and a visual analogue scale (EQ VAS) where individuals rate their health from ‘worst imaginable’ (0) to ‘best imaginable’ (100) health state, [2]. EQ-5D-3L has three severity levels ‘no problem’, ‘some problem’, and ‘extreme problem’ for each dimension which gives 243 health states (3^5^) [3].

Value sets provide weights which transform EQ-5D health states into a summary index. Currently, value sets for the EQ-5D-3L have been developed in more than 30 countries globally and one at a European level [4]. Most of the studies used the time trade-off (TTO) method to derive value sets, followed by the VAS method [2,5]. Studies developing EQ-5D value sets use either a hypothetical or an experience-based perspective. In a hypothetical perspective, participants are asked to imagine specific health states that are described to them [6,7]. In an experience-based perspective, participants provide their valuation of health states usually based on their current experience [8,9,10,11]. It has been shown that the perspective participants take in health state valuations and the methods influence results [12,13,14,15,16,17,18,19].

In EQ-5D-3L valuation studies, with different methods, perspectives, and overall designs, the value sets produced vary from country to country [5,12]. Researchers who have used these value sets to report HRQoL in the different countries have also reported varying findings; differences in method of valuation, as well as perspectives have been cited as reasons for the variations in index [17,18,20]. Studies comparing the Swedish EQ-5D-3L value sets with UK value sets among patients who underwent total hip replacement (THR) and patients with diabetes mellitus, also reported differences in index [21,22]. In the English National Health Services where patient-reported outcomes (PROs) have been used as hospital performance indicators, employing different value sets was shown to lead patients to choosing different hospitals [23].

Transferability of HRQoL indices across jurisdictions is an important issue when looking into the use of different value sets based on varying methods and perspectives in the calculation of index scores [18]. A review which assessed international pharmaco-economics guidelines reported that no clear directions were provided in relation to transferability of quality of life evidence (health state utilities/indices) in most of the countries assessed. The few countries with guidelines provided contrasting recommendations [24]. However, some studies have suggested that transferring value sets to other settings or countries needs to be conducted with caution and have advised against direct application of such value sets. In transferring across settings, adjustment of the value sets to fit the new setting has been suggested [17,18].

Although a number of studies have assessed various value sets as to their comparability [12,13,17,18,20,22], comprehensive investigations of different value sets in a manner that accounts for the potential sources of variation, such as methodology, perspective, and country are still scarce. This is especially true when it comes to specific diseases. The National Quality Registers in Sweden, provide suitable data to conduct such studies owing to their large amount of high quality data [25]. This can provide an opportunity to show the spectrum of problems reported on the EQ-5D dimensions in real world settings by comparing different value sets. Furthermore, data from such registers show the type and proportion of problems reported in the EQ-5D dimensions in routine clinical settings.

The present study aimed at understanding the implications of applying EQ-5D-3L value sets from Sweden, Germany, Denmark, and the UK to derive HRQoL indices for patients undergoing THR in Sweden. The specific objectives were:To determine the concordance among EQ-5D-3L indices based on the value sets and with patient-reported EQ VAS scores compared by valuation methods, perspectives, and countries;To assess the difference among the value sets when analysing longitudinal change of the EQ-5D-3L index compared by valuation methods, perspectives, and countries.

## 2. Methods

### 2.1. The Swedish Hip Arthroplasty Register

This quantitative longitudinal study was conducted in Sweden using data of patients who underwent THR and were recorded in the Swedish Hip Arthroplasty Register (SHAR). It is classified at certification level one status (highest level) due to data quality, inclusion of PROs and use in research among other reasons [26]. The register has been in existence for over four decades and holds data on THR from all clinics in the country and has a 100% coverage of providers performing hip replacement in Sweden [27]. Data on PROs have been recorded since 2002 which includes the EQ-5D-3L questionnaire [28].

### 2.2. Sampling

A total of 128,362 records of THRs were found in SHAR during an 8-year period. Of the records of patients who underwent bilateral THRs, the first one was included in the study in order to ensure independent observations. Patients re-operated within 1 year postoperatively were excluded. Of the remaining 107,715 records, 73,523 complete records on EQ-5D-3L and patient-reported EQ VAS score were included for pre- and 1-year postoperative analyses (Appendix A).

### 2.3. Data

Demographic (age, sex, height and weight), clinical (laterality, i.e., the side operation was conducted, diagnostic indication for surgery, as well as American Society of Anaesthesiologists (ASA) class) and PROs data on patients who underwent THRs were extracted from SHAR [28]. The specific PROs collected pre- and 1 year postoperatively included self-reported health through the EQ-5D-3L instrument, a question on hip pain level and Charnley classes. Self-reported data on the level of hip pain experienced by patients during the past four weeks is provided in five levels: ‘none’, ‘very mild’, ‘mild’, ‘moderate’, and ‘severe’ [28].

Charnley classification of patients is based on self-administered questions and groups patients into one of three groups describing mobility. The groups are: patients with walking impairment due to symptoms from only one hip (Group A); patients with walking impairment due to symptoms from both hips (Group B); and patients with other medical problems affecting their ability to walk (Group C) [29].

### 2.4. Value Sets Compared

A total of eight value sets were employed in the present study. Four were developed using the VAS [8,10,30] and the remaining four were developed using the TTO method [7,10,31,32]. In terms of the perspective taken in the valuation, five of the value sets were developed using a hypothetical perspective [7,30,31,32]. The remaining three value sets employed experience-based perspectives where respondents valued their own health states [8,10]. Specifically, TTO value sets [7,10,31,32] from Sweden, Germany, Denmark, and the UK, as well as VAS value sets [8,10,30] from each of the countries were compared in the study. Regarding perspective, both the TTO and VAS value sets from Denmark and the UK, as well as the TTO value set from Germany, used a hypothetical perspective [7,30,31,32]. On the other hand, the Swedish VAS and TTO and the VAS value set from Germany employed an experience-based perspective [8,10]. Value sets with differing perspectives were not available within one country in the eight compared value sets.

The value sets included in this study were chosen to make comparisons of value sets from countries where experience-based value sets have been developed, starting with Sweden and Germany [8,10]. In order to make comparisons with value sets developed using hypothetical perspectives value sets from Denmark were included [31]. In addition, value sets from the UK were also employed for comparisons purposes considering their wide use [7].

For analyses involving concordance, EQ-5D-3L indices, based on VAS value sets which were reported on a scale of 0 to 100, were divided by 100 to facilitate comparison with the other value sets on a scale with a maximum value of 1 [7,8,10,30,31,32]. A detailed characterisation of the value sets included in the study is provided in the Appendix A.

### 2.5. Data Entry, Analysis, and Interpretation

#### 2.5.1. Descriptive Analysis

Records of patients extracted from SHAR, were assessed for completeness. In cleaning the data, records with complete information on EQ-5D-3L and patient-reported EQ VAS score across the follow-up duration were assessed for extreme or inconsistent values. Prior to the main analysis, EQ-5D-3L indices of patients in SHAR were calculated from the data based on value sets from the preoperative data and 1-year postoperative follow-up (Appendix A). Based on this, descriptive analyses of the data were performed presenting demographic and clinical characteristics of patients through frequency and proportions, as well as means and standard deviations. Similarly, problems reported on the EQ-5D-3L dimensions, EQ VAS score, Charnley classes, and hip pain levels were presented.

#### 2.5.2. Concordance

In the main analyses, concordance among the value sets from the three countries and with observed patient-reported EQ VAS score was assessed using Lin’s concordance correlation coefficient (CCC) test. Lin’s CCC provides a pairwise comparison of EQ-5D-3L indices by assessing concordance. The coefficient ranges from −1 (perfect disagreement) to 1 (perfect concordance). This test was done on the full 243 health states and the data from patient records. Lin’s CCC was used as it provides indication as to how close EQ-5D-3L indices can be when they are calculated using different value sets.

The Spearman’s rank correlation tests, although it does not measure concordance, has been used to provide a context in which findings of Lin’s CCC can be understood by comparing with results of correlation tests [33,34]. Lin’s CCC and Spearman’s rank correlation are different due to the approaches they use in assessing correlation. The Spearman’s rank correlation uses rank instead of values and thus assesses monotonic relationships between two variables [35]. On the other hand, Lin’s CCC evaluates how well a linear relationship with a slope of one and a line passing through the origin (y-intercept = 0) fits, which helps in measuring concordance (how close values of the two variables are to being the same) [34].

In addition, Bland–Altman plots and corresponding limits of agreement between the EQ-5D-3L index were performed (results not shown).

#### 2.5.3. Longitudinal Change in EQ-5D-3L Indices

For assessing the difference in longitudinal change of EQ-5D-3L indices among the value sets compared at the 1-year postoperative follow-up, paired sample *t*-tests were used. Assumptions of a continuous dependent variable, absence of significant outliers, and an approximate normal distribution were checked before conducting the test [36]. In addition, the strength of differences in longitudinal change was assessed through effect size using Cohen’s d [37].

Cohen’s d assesses the strength of differences between mean values of two variables. In doing so, the mean difference is divided by the common or average standard deviation of the two variables, resulting in Cohen’s d value. This helps to express the difference between mean values of the variables in standard deviation units (without the original measurement unit). Cohen’s d of 0.2, 0.5, and 0.8 indicate ‘small’, ‘medium’, and ‘large’ differences in mean values, respectively [37].

All analyses were done using the software R version 3.5.0/3.5.1 [38]. The cut-off point for significance of statistical tests was *p* < 0.05.

## 3. Results

### 3.1. Demographic and Clinical Characteristics of Patients

The mean age of the 73,523 patients included in the study at the preoperative follow-up was 68.2 years. More than half of the patients were categorised in the ASA class II (57.8%). Data on BMI showed that more than two-fifths (42.2%) of the patients were in the overweight category. As to diagnoses leading to THR, primary osteoarthritis accounted for almost all (92.5%) of the cases (Appendix A).

### 3.2. Reported Problems on the PROs Instruments and Mean Scores

Pre- and 1 year postoperatively 160 and 176 health states were recorded, respectively (Appendix A). Preoperatively, more than 90% of patients reported problems in the mobility dimension among both men and women. In addition, the highest proportion of severe problems were reported in the pain/discomfort dimension. The proportions of problems in each dimension decreased 1 year postoperatively (Table 1).

Proportions of patients in the three Charnley classes remained stable 1 year postoperatively. The proportions of women in classes B and C were higher than men both pre- and 1 year postoperatively. Problems reported on the VAS for hip pain showed a decline from more than 83% of patients in the mild or moderate pain categories preoperatively to between 7% and 10% 1 year postoperatively, with higher proportions of problems reported by women (Table 1).

The preoperative mean patient-reported EQ VAS score, 56.0 (SD = 22.3), increased to 76.7 (SD = 19.8) 1 year postoperatively. The mean EQ-5D-3L indices based on all the value sets increased from pre- to 1-year postoperative follow-up. The Swedish TTO value set showed the highest index in both follow-ups and the UK TTO had the lowest index preoperatively and 1 year postoperatively. Among VAS value sets, the Danish VAS showed the lowest indices at both time points (Figure 1).

### 3.3. Concordance among EQ-5D-3L Indices Based on the Value Sets Using the Theoretically Possible 243 Health States

Comparisons of EQ-5D-3L indices based on the value sets among the possible 243 health states, showed Spearman’s rank correlation coefficients ranging from 0.62 to 0.98. On the other hand, the findings of concordance ranged from 0.25 to 0.92 in the comparison based on the 243 health states. The highest levels of concordance (ranging from 0.80 to 0.92) were noted in the comparisons of value sets differing by country only. Comparisons of value sets differing by method showed concordance values ranging from 0.55 to 0.70 (Appendix A).

### 3.4. Concordance among EQ-5D-3L Indices Based on the Value Sets Using Patient Data in SHAR

Comparison of the value sets by method, perspective and country was also performed using the data from patients in SHAR. A range of concordance levels were found in the comparisons by method and by perspective. Value sets differing only by country showed the highest concordance levels. The respective comparisons are presented below.

#### 3.4.1. Comparisons by Method

Lin’s CCC of the comparisons had varying levels of values both at baseline and at 1-year follow-up. At baseline, Lin’s CCC of the three pairwise comparisons of value sets by method of valuation ranged from 0.47 to 0.86. At 1-year follow-up, the values showed an increase in all the three comparisons with Lin’s CCC ranging from 0.65 to 0.96 (Table 2).

#### 3.4.2. Comparisons by Perspective

The comparisons by perspective employed in the valuation included seven pairs of value sets. In addition, these comparisons contained differences by country. Lin’s CCC values showed an increase from baseline to 1-year follow-up. Both at baseline and at 1-year follow-up comparisons involving value sets developed using the VAS valuation method showed higher levels of Lin’s CCC compared to those developed using the TTO valuation method (Table 2).

#### 3.4.3. Comparisons by Country

In the comparisons of the value sets differing by country only, Lin’s CCC ranged from 0.79 to 0.92 at baseline and 0.92 to 0.96 at 1-year follow-up. Comparisons by country showed generally higher levels of Lin’s CCC than the comparisons of value sets differing by methods and perspectives (Table 2).

### 3.5. Comparison of Patient-Reported EQ VAS Score and EQ-5D-3L Indices

A line graph of mean patient-reported EQ VAS score and EQ-5D-3L indices through all the value sets in the study for the 80 most frequent health states is shown in Figure 2 and Appendix A. Among the VAS value sets, indices based on experience-based value sets had more consistent values relative to the patient-reported EQ VAS scores while the Danish and UK VAS showed deviations for a number of health states. Among the TTO value sets, the Swedish TTO value set showed a similar pattern to patient-reported EQ VAS scores preoperatively and 1 year postoperatively, but with consistently higher values. Indices based on the German, Danish, and UK’s TTO value sets had pronounced deviations from the patient-reported EQ VAS score and the VAS value sets for a number of health states both pre- and 1 year postoperatively (Figure 2; Appendix A).

Lin’s CCC of patient-reported EQ VAS scores with EQ-5D-3L indices increased from pre- to 1-year postoperative follow-up. Preoperatively, the highest CCC was observed between the patient-reported EQ VAS score and the Swedish VAS and the Danish TTO value sets. The lowest concordance was seen for the Swedish TTO value set followed by the Danish VAS and UK TTO. One year postoperatively, the Swedish VAS value set had the highest Lin’s CCC with patient-reported EQ VAS score while the Swedish TTO had the lowest followed by the German TTO (Table 3).

### 3.6. Comparison of Value Sets by Longitudinal Change in EQ-5D-3L Index

One year postoperatively, the UK TTO showed the highest increase in EQ-5D-3L index, while the Swedish TTO had the lowest (Appendix A). The differences in the changes of EQ-5D-3L index were statistically significant in all pairwise comparisons. However, their effect sizes (Cohen’s d) had a range of values from 0.02 to 1.17. The difference in EQ-5D-3L index change was among the lowest between value sets differing by country only, particularly for VAS value sets (Table 4).

## 4. Discussion

In this study, concordance among a range of value sets and with patient-reported EQ VAS scores, as well as change in EQ-5D-3L index over time were assessed based on the records of THR patients. The findings showed that, in comparison to value sets differing by method and/or perspective, those differing only by country showed higher levels of concordance in the EQ-5D-3L index. The Swedish VAS value set had the highest concordance with the patient-reported EQ VAS score while the Swedish TTO had the lowest concordance. However, the difference between Swedish TTO and the patient-reported EQ VAS score was of a more consistent pattern (i.e., generally uniform difference with patient-reported EQ VAS score across health states), than that of the other TTO value sets pre- and 1 year postoperatively. Changes in EQ-5D-3L index from preoperative to postoperative follow-up also showed that value sets differing only by country generally showed small differences, particularly for the VAS value sets.

In the comparisons of patient data to the theoretically possible 243 health states, mostly higher levels of concordance were shown in EQ-5D-3L indices calculated using patients’ data. Among possible reasons is the difference in the composition of health states where fewer and milder health states were reported among patients with THR compared to the 243 theoretically possible health states. The proportions of the health states reported could also contribute to the difference, with higher proportions of milder states reported in the patient data while in the analysis of the theoretically possible 243 health states, each health state occurs only once.

No clear pattern was observed in this study concerning concordance in the comparisons involving value sets differing by method, with concordance levels ranging from less than 0.5 to nearly 1.0. One study compared 15 EQ-5D-3L value sets from different countries in terms of coefficients of the models of the value sets and EQ-5D-3L indices for two selected hypothetical health states (a patient with depression and a patient with pain) [18]. Similar to the present study, the findings showed substantial differences among value sets mainly attributed to differences in valuation methods. The study advised against direct transfer of EQ-5D-3L index values across countries due to lack of consistent findings [18]. Similarly, large differences were reported between European and Slovenian VAS value sets and Polish and UK TTO value sets in a study based on a sample of patients with 18 chronic conditions in Hungary [19].

In contrast to the above cited studies, a study among patients with cough or lower respiratory tract infection in seven European countries and another study among the general population in three countries (Germany, The Netherlands, and Spain) reported differences by valuation methods and countries to be small [39,40]. Yet, another study among medical students at a university in China showed similar index values using VAS and TTO valuation methods [41]. The present study showed findings reflective of patterns seen in the cited studies with some of the pairwise comparisons in the present study showing higher concordance while others showed moderate and lower concordance levels. Furthermore, the present study showed that caution needs to be taken in transferring EQ-5D-3L indices calculated using value sets differing by methods.

In our study, comparisons of value sets with different perspectives (with additional difference by country) yielded higher concordance levels for VAS value sets than TTO value sets. A possible reason could be related to differences in the valuation procedures in TTO compared to VAS. The diversity of practical approaches with the TTO method in different valuation studies has been noted in the literature [42]. Similarly, a recent study comparing different TTO value sets in terms of their impact on quality-adjusted life years (QALYs) also showed a range of index values for the value sets from different countries [43]. In addition, the higher concordance, among VAS value sets, than TTO value sets could also be attributed to higher uniformity in the VAS valuation procedures across studies owing to its ease for self-completion [44].

In the literature, the pairs of value sets varying only by country demonstrated higher concordance levels than other comparisons. In line with the present study, more similarities in value sets with the same method and perspective were shown in two studies comparing VAS value sets (Iran and UK) and TTO value sets (Brazil and UK). One is based on the theoretically possible 243 EQ-5D-3L health states [21], while the study from Brazil used data of multiple sclerosis patients from eight sites [45]. These studies showed that levels of agreement were different for mild and severe health states [21,45]. In line with the cited studies, the present study showed that differences by country only could permit transferability of value sets across countries better than those differing by methods and perspectives.

In contrast, other studies reported substantial variations between value sets across countries [46,47,48]. One of these studies assessed TTO value sets, with comparisons between Argentina, Chile, and the UK, where EQ-5D-3L health states obtained through a survey using hypothetical description of pneumococcal and human papilloma virus diseases were employed [46]. Another study was based on data of Crohn’s patients in Italy using TTO values sets from Italy, the UK, and USA [47]. In yet another study, data of rheumatoid arthritis patients and the theoretical 243 health states were compared using the TTO value sets from Denmark, the UK, and USA [48]. All the comparisons in the cited studies involve TTO value sets and variation in the elicitation procedures in TTO valuation could be one of the possible reasons for differences, as noted above [42].

In addition, a review which compared TTO valuation studies reported variations in value sets among different countries. It also showed larger differences between the UK value set and non-northern European countries compared to northern European countries [49]. In the present study, most of the countries compared were from northern Europe, which are similar in many sociocultural aspects. This could be a further reason for the high concordance between value sets different by country only. Cultural differences have been suggested among possible reasons for variations in EQ-5D value sets [49].

In all the pairwise comparisons among value sets and with patient-reported EQ VAS scores, the level of concordance among indices from the value sets were higher postoperatively than preoperatively. This could be due to the difference in the proportion of health states at the two time points, where full health and other milder health states were more common postoperatively than preoperatively.

The lower concordance between patient-reported EQ VAS score and EQ-5D-3L indices compared to analyses of value sets indicates a systematic difference between patient-reported EQ VAS scores and EQ-5D-3L indices from the value sets. Value sets render a single value per health state but were shown to not fully capture the information provided by patient-reported EQ VAS score leading to systematic difference [50]. Similarly, lower levels of concordance were reported by a study which compared patient-reported EQ VAS score with value sets from Malaysia, Singapore, Thailand, and the UK [51].

As to the longitudinal change in EQ-5D-3L index, the UK TTO showed the highest increase while the Swedish TTO recorded the smallest increase. This could partly be attributed to the difference between the value sets in the valuation of severe and milder health states, noted above. The UK and other hypothetical TTO value sets showed high indices (close to the maximum of 1) for milder health states while yielding much lower indices (0 and negative values) for severe health states, especially the UK value set with about one third of the health states were worse than being dead. This contributed to a larger increase in index from pre- to 1-year postoperative follow-up which was further pronounced by the presence of health states with indices below 0 in the hypothetical TTO value sets. In addition, the hypothetical TTO value sets have the highest decrements in the coefficients in the mobility and pain dimensions, where high proportions of problems were reported in our population preoperatively [7,31,32]. In contrast, the Swedish TTO showed a generally consistent high index across health states from mild to severe ones with a value of 0.34 for health state ‘33333’ [10]. A study in Sweden which compared the Swedish TTO with the UK hypothetical TTO, with one third proportion of negative health states, value set reported similar findings to the present study stating that the Swedish TTO had consistently high indices and smaller changes longitudinally [14].

In the present study, there are several limitations to be taken into consideration. Due to value sets available, comparisons involving perspectives could only be pursued across different countries. This prevented direct comparison between perspectives that could possibly affect the findings. Though this prevented separating the influence of these two factors, difference by country did not have a major impact in the other comparisons. Another issue which cannot be fully explored in our study is the different time points of the valuation studies for the value sets. The UK value sets were based on data collected in 1993. The German and Danish TTO as well as the Danish VAS value sets were all produced in the late 1990s and early 2000s. The rest of the value sets from Sweden and Germany were developed based on data from the late 2000s and early 2010s [7,8,10,30,31,32]. In addition, among the compared value sets, in the development of the Swedish and German VAS value sets, anchoring for dead based on the state ‘dead’ was not done. Hence, the EQ-5D-3L indices from the two value sets were not rescaled when employed in the present study.

For THR patients, the present study adds to the current discussion on the comparison of value sets for the EQ-5D-3L by assessing value sets that differ with respect to method, perspective, as well as country. Our study findings provide information on which factors to take into account when transferring HRQoL and derived utility values from one setting to another. Results help to inform resource allocation decisions, based on QALYs, and decisions considering HRQoL as a medical endpoint based on non-preference based valuations. A recent study from the UK also showed the real-world impact of information on the performance of health institutions on patients’ choice of health institutions [23].

In addition, our findings on concordance among different value sets in terms of method, perspective and country differences helps inform the need for applying crosswalks between value sets through clearer characterisation of differences among value sets. A cross walk is relevant in cases where direct transfer of value sets is not meaningful and an approach that facilitates transfer of EQ-5D indices between value sets has been reported in a previous study [52].

## 5. Conclusions

In the present study, on THR patients, the findings indicate that when the method and perspective used in the value sets are similar, concordance between EQ-5D-3L indices remained high despite differences in countries where the value sets were developed. Given the high concordance, the value sets could be adopted across the respective settings. In addition, in cases of differences in methods and perspectives among value sets, transfer of value sets across settings should consider conversion through crosswalk. More generally, it is recommended that obtaining information on value sets in terms of the country, perspective, and method employed in their development is crucial before transferring HRQoL and utility index data across countries.

## Figures and Tables

**Figure 1 jcm-10-04205-f001:**
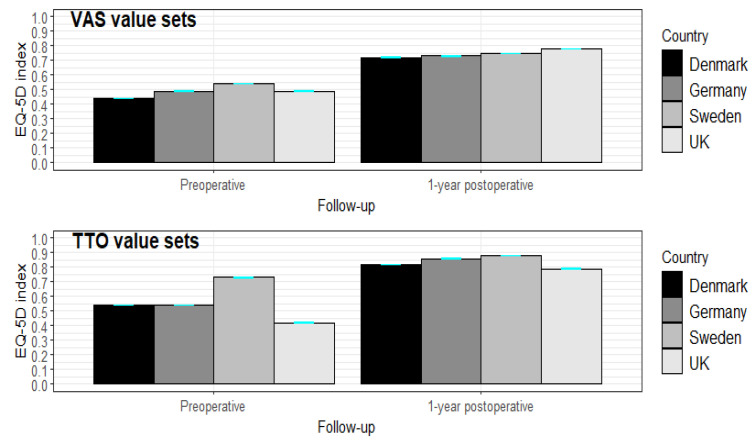
Mean EQ-5D-3L indices (95% CI) among VAS and TTO value sets among THR patients by follow-up (preoperative and 1-year postoperative) (*n* = 73,523).

**Figure 2 jcm-10-04205-f002:**
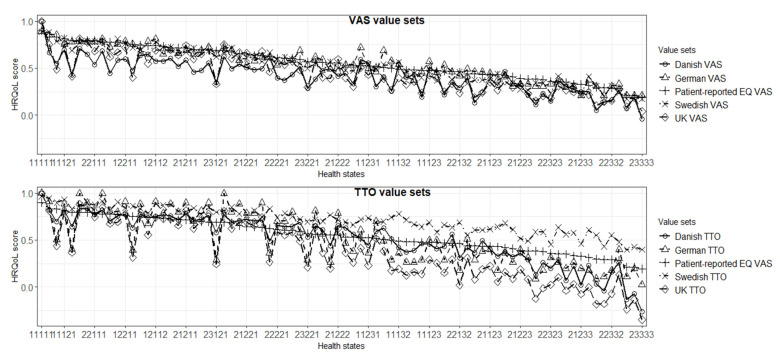
Comparison of patient-reported EQ VAS score with EQ-5D-3L indices based on the value sets 1 year postoperatively (ordered by patient-reported EQ VAS) (*n* = 73,523).

**Table 1 jcm-10-04205-t001:** Prevalence of reported problems in the EQ-5D-3L dimensions, Charnley classes, and hip pain level by sex pre- and 1 year postoperatively (*n* = 73,523).

Variable	Follow-Up							
	Preoperative				1-Year Postoperative			
	Men		Women		Men		Women	
	%	*n*	%	*n*	%	*n*	%	*n*
**EQ-5D-3L dimension**								
**Mobility**								
No problem	8.9	2825	6.9	2878	67.7	21,507	55.7	23,276
Some problems	90.9	28,856	92.7	38,733	32.2	10,212	44.2	18,442
Severe problems	0.2	73	0.4	158	0.1	35	0.1	51
**Self-care**								
No problem	76.9	24,422	77.5	32,361	93.2	29,583	92.4	38,586
Some problems	22.3	7081	21.5	9000	6.3	2007	7.0	2943
Severe problems	0.8	251	1.0	408	0.5	164	0.6	240
**Usual activities**								
No problem	40.4	12,842	38.1	15,914	81.2	25,793	75.5	31,541
Some problems	49.2	15,634	52.2	21,784	17.0	5409	22.5	9415
Severe problems	10.3	3274	9.7	4071	1.7	552	1.9	813
**Pain/discomfort**								
No problem	1.7	539	1.4	567	51.2	16,255	39.6	16,534
Some problems	64.7	20,543	51.0	21,287	45.4	14,426	55.0	22,970
Severe problems	33.6	10,672	47.7	19,915	3.4	1073	5.4	2265
**Anxiety/depression**								
No problem	65.9	20,934	52.8	22,055	83.7	26,565	75.0	31,306
Some problems	31.8	10,094	43.0	17,961	15.2	4839	23.4	9768
Severe problems	2.3	726	4.2	1753	1.1	350	1.7	695
**Charnley class**								
Category A	53.1	16,846	43.3	18,074	53.7	17,055	42.5	17,768
Category B	12.1	3837	13.0	5417	10.4	3298	11.1	4653
Category C	34.9	11,071	43.8	18,278	35.9	11,400	46.3	19,345
Missing	0.0	0	0.0	0	0.0	1	0.0	3
**Hip pain level**								
None	1.9	608	1.1	465	80.5	25,571	75.9	31,682
Very mild	10.0	3186	5.5	2316	11.7	3708	14.2	5936
Mild	38.9	12,355	31.1	12,975	5.6	1776	7.2	3015
Moderate	44.6	14,172	52.8	22,051	2.0	641	2.5	1039
Severe	4.5	1433	9.5	3962	0.2	57	0.2	94
Missing	0.0	0	0.0	0	0.0	1	0.0	3

**Table 2 jcm-10-04205-t002:** Lin’s CCC test between EQ-5D-3L indices based on the value sets pre- and 1 year postoperatively (*n* = 73,523).

Comparison Approach	Follow-Up			
	Preoperative		1-Year Postoperative	
	Lin’s CCC	95% CI	Lin’s CCC	95% CI
**Method**				
Swedish VAS vs. Swedish TTO	0.47	[0.47, 0.48]	0.65	[0.64, 0.65]
Danish VAS vs. Danish TTO	0.65	[0.65, 0.66]	0.80	[0.80, 0.80]
UK VAS vs. UK TTO	0.86	[0.86, 0.86]	0.96	[0.96, 0.96]
**Perspective**				
Swedish VAS vs. Danish VAS	0.73	[0.73, 0.73]	0.82	[0.81, 0.82]
Swedish VAS vs. UK VAS	0.85	[0.85, 0.86]	0.89	[0.89, 0.89]
German VAS vs. Danish VAS	0.83	[0.83, 0.84]	0.89	[0.89, 0.89]
German VAS vs. UK VAS	0.86	[0.86, 0.86]	0.89	[0.89, 0.89]
Swedish TTO vs. German TTO	0.40	[0.40, 0.41]	0.76	[0.76, 0.76]
Swedish TTO vs. Danish TTO	0.47	[0.47, 0.48]	0.76	[0.76, 0.76]
Swedish TTO vs. UK TTO	0.30	[0.29, 0.30]	0.63	[0.63, 0.63]
**Country**				
Swedish VAS vs. German VAS	0.92	[0.92, 0.93]	0.96	[0.96, 0.96]
Danish VAS vs. UK VAS	0.79	[0.78, 0.79]	0.92	[0.92, 0.92]
German TTO vs. Danish TTO	0.92	[0.92, 0.93]	0.92	[0.92, 0.93]
German TTO vs. UK TTO	0.90	[0.90, 0.90]	0.90	[0.90, 0.90]
Danish TTO vs. UK TTO	0.84	[0.84, 0.84]	0.95	[0.95, 0.95]

**Table 3 jcm-10-04205-t003:** Lin’s CCC tests between patient-reported EQ-VAS and EQ-5D-3L indices based on the value sets pre- and 1 year postoperatively (*n* = 73,523).

Comparison	Follow-Up			
	Preoperative		1-Year Postoperative	
	Lin’s CCC	95% CI	Lin’s CCC	95% CI
**Patient-reported EQ VAS vs.**				
Swedish VAS	0.43	[0.42, 0.43]	0.70	[0.69, 0.70]
German VAS	0.38	[0.37, 0.38]	0.68	[0.68, 0.69]
Danish VAS ^a^	0.33	[0.32, 0.33]	0.65	[0.64, 0.65]
UK VAS ^a^	0.40	[0.40, 0.41]	0.69	[0.69, 0.70]
Swedish TTO	0.25	[0.25, 0.26]	0.48	[0.48, 0.48]
German TTO ^a^	0.37	[0.36, 0.37]	0.58	[0.57, 0.58]
Danish TTO ^a^	0.43	[0.43, 0.44]	0.67	[0.66, 0.67]
UK TTO ^a^	0.34	[0.34, 0.35]	0.67	[0.67, 0.68]

^a^ Hypothetical perspective.

**Table 4 jcm-10-04205-t004:** Paired-sample *t* test of mean difference in change of EQ-5D-3L indices between value sets 1 year postoperatively (n= 73,523).

Value Sets	Mean of Difference in Change	[95%CI]	Effect Size (Cohen’s d)
**Method**			
Swedish VAS vs. Swedish TTO	0.066 *	[0.065, 0.066]	1.17
Danish VAS vs. Danish TTO	0.003 *	[0.002, 0.005]	0.02
UK VAS vs. UK TTO	−0.083 *	[−0.084, −0.082]	−0.65
**Perspective**			
Swedish VAS vs. Danish VAS	−0.068 *	[−0.068, −0.067]	−0.50
Swedish VAS vs. UK VAS	−0.077 *	[−0.078, −0.076]	−0.69
German VAS vs. Danish VAS	−0.045 *	[−0.046, −0.044]	−0.40
German VAS vs. UK VAS	−0.055 *	[−0.056, −0.054]	−0.51
Swedish TTO vs. Danish TTO	−0.130 *	[−0.131, −0.129]	−0.88
Swedish TTO vs. German TTO	−0.168 *	[−0.169, −0.166]	−0.78
Swedish TTO vs. UK TTO	−0.226 *	[−0.227, −0.224]	−0.94
**Country**			
Swedish VAS vs. German VAS	−0.022 *	[−0.023, −0.022]	−0.46
Danish VAS vs. UK VAS	−0.011 *	[−0.011, −0.009]	−0.08
German TTO vs. Danish TTO	0.038 *	[0.037, 0.039]	0.34
German TTO vs. UK TTO	−0.058 *	[−0.058, −0.057]	−0.78
Danish TTO vs. UK TTO	−0.096 *	[−0.096, −0.095]	−0.88

* *p*-value < 0.001.

## Data Availability

The data underlying the findings of the study are not publicly available due to the requirement of confidentiality under which the study was approved.

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
