# Peer review of "Concordance among Swedish, German, Danish, and UK EQ-5D-3L Value Sets: Analyses of Patient-Reported Outcomes in the Swedish Hip Arthroplasty Register"

_jcm, 2021, doi:10.3390/jcm10184205_

Round 1
Reviewer 1 Report
-
in the manuscript 2 additional file are mentioned. However, there is only one additional file available, which include tables S1-5 and Figures 1-3 together
-
check the naming of the figures in the additional file. There is a mixture of different abbreviations: Figure 1, Figure S2 and Figure 3 (in the manuscript the numbering of all additional figures include a "S")
-
Supplementary - table S1: include "ordinary least square" in the last line of the table, instead of "OLS"
Author Response
Thank you for your positive assessment of our manuscript. We have below provided replies to the specific points raised. Changes are shown in track change in the manuscript and additional file.
- In the manuscript 2 additional file are mentioned. However, there is only one additional file available, which include tables S1-5and Figures 1-3 together
- - Thank you. We have updated the mention of two files in the manuscript text to be just ‘Additional file’ as all additional tables and figures are provided in one additional file.
- Check the naming of the figures in the additional file. There is a mixture of different abbreviations: Figure 1, Figure S2 and Figure3 (in the manuscript the numbering of all additional figures include a "S")
- - Thank you. We have now indicated the specified figure in the additional file as “Figure S3”
- Supplementary - table S1: include "ordinary least square" in the last line of the table, instead of "OLS"
- - Thank you. We have now replaced OLS by ordinary least squares in the last row of Additional file, Table S1
Reviewer 2 Report
well written useful paper
Author Response
Thank you for your positive assessment of our manuscript
- We have performed checks on the English and the style of the manuscript and it is marked by track change throughout the manuscript
Reviewer 3 Report
- I would recommend putting error bars (95%CI) in figure 1, though the sample size was quite big and the 95%CI might be very small.
Author Response
Thank you for your positive assessment of our manuscript. We have addressed your comment as detailed below under the comments
- I would recommend putting error bars (95%CI) in figure 1, though the sample size was quite big and the 95%CI might be very small.
- - Thank you, we have now added a 95%CI marker on the bars in Figure 1 shown in Turquoise color.
Reviewer 4 Report
Dear authors, thank you for allowing me to read your interesting work.
This is a nice paper with massive numbers of patients about concordance among different countries EQ-5D-3L value sets.
I wish to know the data of other species, including American, Asian, and African. However, this is a future challenges. So, this paper is valuable for publishing in Journal of Clinical Medicine.
Author Response
Thank you for your positive assessment of our manuscript. We have provided a reply to your comment below under the comment.
- This is a nice paper with massive numbers of patients about concordance among different countries EQ-5D-3L value sets. I wish to know the data of other species, including American, Asian, and African. However, this is a future challenges. So, this paper is valuable for publishing in Journal of Clinical Medicine.
- - Thank you. We hope to see similar studies in specified different settings as well.